

# Strategy to improve the accuracy of convolutional neural network architectures applied to digital image steganalysis in the spatial domain

Reinel Tabares-Soto[1], Harold Brayan Arteaga-Arteaga[1], Alejandro Mora-Rubio[1], Mario Alejandro Bravo-Ortíz[1], Daniel Arias-Garzón[1], Jesús Alejandro Alzate Grisales[1], Alejandro Burbano Jacome[1], Simon Orozco-Arias[2,3], Gustavo Isaza[3] and Raul Ramos Pollan[4]

[1] Department of Electronics and Automation, Universidad Autónoma de Manizales, Manizales, Caldas, Colombia
[2] Department of Computer Science, Universidad Autónoma de Manizales, Manizales, Caldas, Colombia
[3] Department of Systems and Informatics, Universidad de Caldas, Manizales, Caldas, Colombia
[4] Department of Systems Engineering, Universidad de Antioquia, Medellín, Antioquia, Colombia

## ABSTRACT

In recent years, Deep Learning techniques applied to steganalysis have surpassed the traditional two-stage approach by unifying feature extraction and classification in a single model, the Convolutional Neural Network (CNN). Several CNN architectures have been proposed to solve this task, improving steganographic images' detection accuracy, but it is unclear which computational elements are relevant. Here we present a strategy to improve accuracy, convergence, and stability during training. The strategy involves a preprocessing stage with Spatial Rich Models filters, Spatial Dropout, Absolute Value layer, and Batch Normalization. Using the strategy improves the performance of three steganalysis CNNs and two image classification CNNs by enhancing the accuracy from 2% up to 10% while reducing the training time to less than 6 h and improving the networks' stability.

## INTRODUCTION

Steganography and steganalysis are two research fields related to hidden messages in digital multimedia files. The first one consists of hiding information and the second one on detecting whether a file has a message or not. The steganographic process is illustrated by the famous Prisoner Problem (*Simmons, 1984*), which presents a prison scenario where two prisoners try to exchange messages reviewed by the prison director, who decides whether to deliver the message content. Steganography referring to images can be applied in the spatial or frequency domain. In the spatial field, the algorithms change the value of some pixels on the image, making imperceptible to the human eye, for instance, the Least Significant Bits (LSB) of each pixel (*Johnson & Jajodia, 1998*; *Fridrich, Goljan & Du, 2001*). Some of the algorithms in this domain are HUGO (*Pevny, Filler & Bas, 2010*), HILL (*Li et al., 2014*), MiPOD (*Sedighi et al., 2016*), S-UNIWARD (*Holub, Fridrich &*

Corresponding author
Reinel Tabares-Soto,
rtabares@autonoma.edu.co

*Denemark, 2014*), and WOW (*Holub & Fridrich, 2012*). On the frequency domain, steganography use several transformations, such as Discrete Cosine Transform (DCT), Discrete Wavelet Transform (DWT), and Singular Value Decomposition (SVD) (*Tabares-Soto et al., 2020*). The compression format Joint Photographic Experts Group (JPEG) is the most common and based on DCT, whose coefficients can be modified to include a hidden message. Some of the algorithms in this domain are J-UNIWARD (*Holub, Fridrich & Denemark, 2014*), F5 (*Westfeld, 2001*), UED (*Guo, Ni & Shi, 2014*), and UERD30 (*Guo et al., 2015*).

Steganalysis consists of detecting whether an image has a hidden message, is divided into two stages. Stage one consists of feature extraction (e.g., Rich Models (RM) by *Fridrich & Kodovsky (2012)*), step two consists of binary classification (an image has a hidden message or not) where are typically used models such as Support Vector Machines (SVM) or perceptrons. In more recent years, thanks to advances in Deep Learning (DL) (*Theodoridis, 2015*) and Graphics Processing Unit (GPUs) (*Tabares Soto, 2016*), DL techniques in steganography and steganalysis have been improving the detection percentages of steganographic images. These techniques are characterized by the unification of feature extraction stage and classification under the same model, reducing complexity and dimensionality introduced by manual feature extraction (*Fridrich & Kodovsky, 2012*).

*Qian et al. (2015)* designed the first Convolutional Neural Network (CNN) with a supervised learning approach. This CNN consisted of convolutional layers and used *Gaussian Activation*. Compared to the detection percentages of state-of-the-art approaches; the results were approximately 4% lower than those obtained by Spatial Rich Models (SRM) *Fridrich & Kodovsky (2012)*, and approximately 10% higher than those obtained by Subtractive Pixel Adjacency Matrix (SPAM) *Pevny, Bas & Fridrich (2010)*.

*Xu, Wu & Shi (2016b)* proposed an architecture with convolutional layers, similar to *Qian et al. (2015)*. The researchers improved the detection percentages by using an absolute value (ABS) layer and $1 \times 1$ convolutional kernels and modifying the training scheme using CNN as a Base Learner (*Xu, Wu & Shi, 2016a*) train sets of CNNs, obtaining better training parameters. In the same year, *Qian et al. (2016)* used Transfer Learning, for the first time in the field, by taking the trained parameters of a CNN with high payload steganographic images and using them to detect images with a low payload. At this point, DL techniques still would not surpass SRM nor SPAM.

*Ye, Ni & Yi (2017)* presented a new CNN with eight convolutional layers, Truncation Linear Unit (TLU) as activation function, and filter banks initialized with SRM-based weights for image preprocessing. By mimicking the SRM feature extraction process, the detection results were improved by approximately 10% compared to traditional algorithms.

*Yedroudj, Comby & Chaumont (2018)* proposed a new CNN. This CNN joined different concepts that were useful in previous designs: filter banks based on SRM, five convolutional layers for feature extraction, Batch Normalization (BN), TLU activation units, and a more significant training database, achieved by adding images from the *BOWS 2* database (*Mazurczyk & Wendzel, 2017*) to the traditional *BOSSBase* database (*Bas, Filler*

*& Pevny, 2011*), and also by crop, resize, rotate and interpolate operations. With this new architecture, the detection results were improved by approximately 6% compared to *Ye, Ni & Yi (2017)*. In the same year, *Boroumand, Chen & Fridrich (2018)* proposed a CNN architecture that detects steganographic images in the spatial and frequency domains. This new architecture used filter banks based on SRM optimized during training, as well as shortcuts or residual connections to allow training such a deep network.

*Zhang et al. (2019)* proposed a new architecture. This CNN uses an SRM-inspired filter bank in the preprocessing layer weights, separable convolutions, and multi-level average pooling known as Spatial Pyramid Pooling (SPP) (*He et al., 2014*) to allow the network to analyze arbitrarily sized images.

The most recent contribution was published by *Reinel et al. (2021)*. This network maintains, for the preprocessing stage, the 30 SRM filters and has a $3 \times$ TanH activation function. This CNN uses shortcuts for feature extraction and separable and depthwise convolutions. Also, the architecture uses the ELU activation function on all feature extraction convolutions. The CNN does not use fully connected layers; the network uses a softmax directly after the global average pooling. Up until now, this CNN achieves the best detection percentage of steganographic images in the spatial domain.

This article presents a thorough experimentation process where different CNNs architectures were tested under various combinations of computational elements and hyper-parameters to determine which of them are relevant in steganalysis, and then design a strategy to improve steganographic image detection accuracy for multiple architectures. The strategy makes modifications in each stage of the network: preprocessing, feature extraction, and classification. The proposed changes improved the accuracy of three steganalysis CNNs from 2% up to 10% while reducing the training time to less than 6 h and enhancing the stability of the networks. Additionally, this approach allows us to adapt image classification architectures (e.g., VGG16 or VGG19) to the steganalysis application. It is noteworthy the SRM filters for preprocess information and improves accuracy, as well as Spatial Dropout for training stability. Still, all layers are essential to assess the relevance of different aspects of DL algorithms applied to steganalysis, ultimately helping to understand the limitations and approach them.

The rest of the paper has the following order: "Materials and Methods" describes the database, computational elements involved in the strategy, and the CNNs architectures engaged in the experiments. "Results" present the results found. "Discussion" analyses and discusses the results. At last, "Conclusions" presents the conclusions of the paper.

# MATERIALS AND METHODS

## Databases

The databases used for the experiments were *Break Our Steganographic System* (BOSSBase 1.01) (*Bas, Filler & Pevny, 2011*) and *Break Our Watermarking System* (BOWS 2) (*Mazurczyk & Wendzel, 2017*). These databases are frequently applied for steganalysis in the spatial domain. Each database has 10,000 cover images in a Portable Gray Map (PGM) format, $512 \times 512$ pixels, and bits in grayscale. BOSSBase and BOWS 2 have similar features and capture devices to avoid the Cover-Source Mismatch effect (*Kodovský,*

*Sedighi & Fridrich, 2014*; *Pibre et al., 2016*; *Chen et al., 2017*). For this research, we established a baseline for all the experiments, the following operations were performed on the images:

- All images were resized to 256 × 256 pixels.
- Each corresponding steganographic image was created for each cover image using two different algorithms, two payloads of 0.2 bits per pixel (bpp) and 0.4 bpp.
- The images were divided into training, validation, and test sets, creating two databases. One with images from BOSSBase 1.01 and the other combining BOSSBase 1.01 and BOWS 2.
- • Each set was saved in *NumPy array* (npy) format, which decreases reading time from 16 to 20 times.

### *Partition*

We used two database for the experiments, BOSSBase 1.01 and BOSSBase 1.01 + BOWS 2. The BOSSBase 1.01 database contains 10,000 pairs of images (cover and stego) divided into 4,000 pairs for training, 1,000 pairs for validation, and 5,000 for testing. The partition of the BOSSBase 1.01 database was based on the works by *Xu, Wu & Shi (2016b)*, *Ye, Ni & Yi (2017)* and *Zhang et al. (2019)*.

The BOSSBase 1.01 + BOWS 2 database contains 20,000 pairs of images, divided into 14,000 pairs for training (10,000 BOWS 2 + 4,000 BOSSBase 1.01), 1,000 pairs for validation (BOSSBase 1.01) and 5,000 for testing (BOSSBase 1.01). The distribution and partition for this database was done as proposed by *Ye, Ni & Yi (2017)*, *Yedroudj, Comby & Chaumont (2018)* and *Zhang et al. (2019)*.

## Steganographic algorithms

Two steganographic algorithms were used to embed noise in the cover images from the databases; these were: S-UNIWARD by *Holub, Fridrich & Denemark (2014)* and WOW by *Holub & Fridrich (2012)* with two payloads (0.2 and 0.4 bpp). The steganographic algorithms based implementation was on the open-source tool named Aletheia (*Lerch, 2020*) and open-source implementation by Digital Data Embedding Laboratory at Binghamton University (*BinghamtonUniversity, 2015*).

## Computational elements

### *SRM filter banks*

SRM filters were designed by *Fridrich & Kodovsky (2012)* to enhance and extract steganographic noise from images. These filters were designed and used in steganalysis before introducing CNNs to the field, but as shown by *Ye, Ni & Yi (2017)* and *Yedroudj, Comby & Chaumont (2018)*, using these filters to initialize the kernels of a convolutional layer improves detection results. Inspired by these works, the preprocessing block uses 30 high-pass filters from the SRM before the feature extraction stage, the selected filters are presented on Fig. 1. It is important to note that the convolution kernels' size was set to 5 × 5 and to achieve that, some of the filters were padded with zero.

**Figure 1** SRM filters per categories.

### Batch normalization

Batch normalization (BN) consists of normalizing each feature distribution, making the average zero and the variance unitary, which results in less sensitive training to the initialization of parameters. This operation allows scaling and translating the distribution, if necessary (*Ioffe & Szegedy, 2015*). In practice, BN allows for a higher learning rate and improves detection accuracy (*Chen et al., 2017*). Eq. (1) describes the BN used in this study.

Given a random variable $X$ whose realization is a value $x \in \mathbb{R}$ of the feature map, the BN of this value $x$ (*Reinel, Raul & Gustavo, 2019*; *Tabares-Soto et al., 2020*) is:

$$BN(x, \gamma, \beta) = \beta + \gamma \frac{x - E[X]}{\sqrt{Var[X] + \varepsilon}} \tag{1}$$

with $E[X]$ the expectation, $Var[X]$ the variance, and $\gamma$ and $\beta$ two scalars represent a re-scaling and a re-translation. The expectation and the variance are computed per batch, while $\gamma$ and $\beta$ are optimized during training.

### Absolute value layer

An ABS layer computes the absolute value of the feature maps. When applied in steganalysis, it forces the statistical modeling to take the symmetry of noise residuals into account (*Xu, Wu & Shi, 2016b*).

### Spatial dropout

Spatial Dropout was introduced by *Tompson et al. (2015)* as a type of Dropout for CNN, which improves generalization and reduces overfitting. Compared to traditional Dropout, which "drops" the neuron's activation, Spatial Dropout "drops" the entire feature map.

### Truncated linear unit activation function

The Truncated linear unit (TLU) activation function was first introduced by *Ye, Ni & Yi (2017)* as a steganalysis particular activation function. This function's motivation is to capture the external signal to noise ratio characteristic of the steganographic embedding procedure, which in general, embeds signals with a much lower amplitude than image content. TLU is a modification of the joint Rectified Linear Unit (ReLU), which performs linear activation, but is truncated at threshold $T$. TLU is defined in Eq. (2).

$$TLU(x) = \begin{cases} -T & \text{if } x < -T \\ x & \text{if } -T \leq x \leq T \\ T & \text{if } x > T \end{cases} \tag{2}$$

### Leaky rectified linear unit activation function

Leaky Rectified Linear Unit (ReLU) is another modification of the ReLU activation function. In this case, for negative values, the function decreases linearly controlled by the negative slope $m$. This slight modification is useful for specific applications, given it avoids the potential problem of a neuron's output always being zero, the Leaky ReLU is less sensitive to weight initialization and data normalization. Equation (3) defines this activation function.

$$LeakyReLU(x) = \begin{cases} mx & \text{if } x < 0 \\ x & \text{if } x \geq 0 \end{cases} \tag{3}$$

### Hyperbolic tangent activation function

Hyperbolic Tangent (TanH) activation function is commonly used in neural networks. It provides nonlinear activation while being a smooth differentiable function. TanH range is also constrained. Equation (4) defines TanH.

$$\text{TanH}(x) = \frac{e^x - e^{-x}}{e^x + e^{-x}} \tag{4}$$

## CNN architectures

The strategy was developed and tested on three CNN architectures designed for steganalysis in the spatial domain and two image classification CNN architectures.

### Xu-Net

Xu-Net is the name for the CNN proposed by *Xu, Wu & Shi (2016b)*. This architecture has a feature extraction stage composed of a High Pass Filter (HPF) layer, five convolutional layers for feature extraction, an ABS layer after the first convolutional layer, BN after each convolutional layer, a classification stage that consists of two fully connected layers, and one Softmax. The first two layers use the TanH activation function and ReLU for the last three layers (*Reinel, Raul & Gustavo, 2019*).

The mini-batch gradient descent optimizer was used for the training process, with momentum fixed to 0.9, and the learning rate initialized to 0.001, scheduled to decrease 10% every 5,000 iterations. Each mini-batch consisted of 64 images (32 cover/stego pairs). The CNN was trained for 120,000 iterations.

### Ye-Net

This network proposed by *Ye, Ni & Yi (2017)*, it uses an SRM filter bank for steganographic noise extraction. The feature extraction stage consists of eight convolutional layers, a TLU activation function after the first layer, and TanH for the others. The classification stage has one fully connected layer and Softmax activation function.

In the original Ye-Net work, the AdaDelta optimizer was used, with momentum fixed to 0.95, weight decay set to $5 \times 10^{-4}$, the "delta" parameter was $1 \times 10^{-8}$, and the learning rate initialized to 0.4. Each mini-batch consisted of 32 images (16 cover/stego pairs). The CNN was trained for different number of epochs based on the experiment and behavior of accuracy.

### Yedroudj-Net

This network proposed by *Yedroudj, Comby & Chaumont (2018)*, it takes the best features of the Xu-Net, and Ye-Net and unifies them under the same architecture. This architecture uses an SRM-inspired filter bank, five convolutional layers for feature extraction, an ABS layer after the first one, Average Pooling after each layer, starting from the second one. It uses the TLU activation function in the first two layers and ReLU in the last three layers. The classification stage has two fully connected layers and Softmax activation function.

They applied a mini-batch stochastic gradient descent (SGD) optimizer. The momentum was fixed to 0.95 and the weight decay to 0.0001. The learning rate (initialized to 0.01) was decreased by a factor of 0.1, each 10% of the total number of epochs. The mini-batch size was set to 16 (8 cover/stego pairs), due to GPU memory limitation.

### VGG16 and VGG19

VGG16 and VGG19 are CNNs proposed by *Simonyan & Zisserman (2015)*. These architectures were initially designed for image classification and presented for the Large Scale Visual Recognition Challenge 2014 (*StanfordVisionLab, 2014*), achieving 93.2% top-5 test accuracy in ImageNet. The number in the network name represents the number of weight layers each architecture has; VGG16 has 16 weight layers (13 convolutional and three fully connected layers). VGG19 has 19 weight layers (16 convolutional and three fully connected layers). Both architectures consist of 5 convolutional blocks (variable number of convolutional layers), each followed by Max or Average Pooling, three fully connected layers, and Softmax activation function at the end for classification purposes. All hidden layers use the ReLU activation function.

## Strategy

With all the computational elements mentioned before, we transform all architectures by the following changes:

- Input image resized to 256 × 256
- All SRM filters were applied in the preprocessing block by a convolution, followed by a 3 × TanH activation, which is a modified TanH with range (−3,3).
- Spatial Dropout applied in Convolutional blocks beginning with the second one.
- Activation use in Convolutional blocks were Leaky ReLU.
- Add Absolute layer (ABS) after activation in Convolutional blocks.
- Batch Normalization layer (BN) after the absolute layer in Convolutional blocks.
- Concatenation layer with triple input of the last layer, located after the first and last BN.
- The classification stage, shown in Fig. 2, consists of three fully connected layers (128, 64 and 32 units, respectively) with Leaky ReLU activation and Softmax activation function. This stage is located after the global average pooling layer and is the same in all architectures.
- The optimizer was stochastic gradient descent.

Figure 3 presents the changes in Ye-Net Architecture as an example of how the strategy is applied, and Fig. 4 shows strategy applied over classification models (Specifically VGG16).

## Hyper-parameters

Convolutional and fully connected layers weights have a glorot normal initializer and use L2 regularization for kernels and bias. The spatial dropout rate has an 0.1 value. BN has a momentum of 0.2, epsilon of 0.001, and renorm momentum of 0.4 value. The stochastic

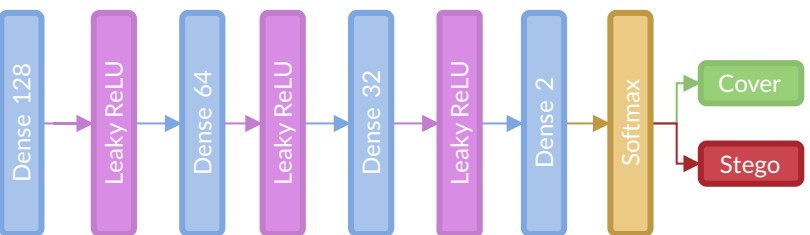

**Figure 2 Fully connected or classification stage implemented for the strategy.**

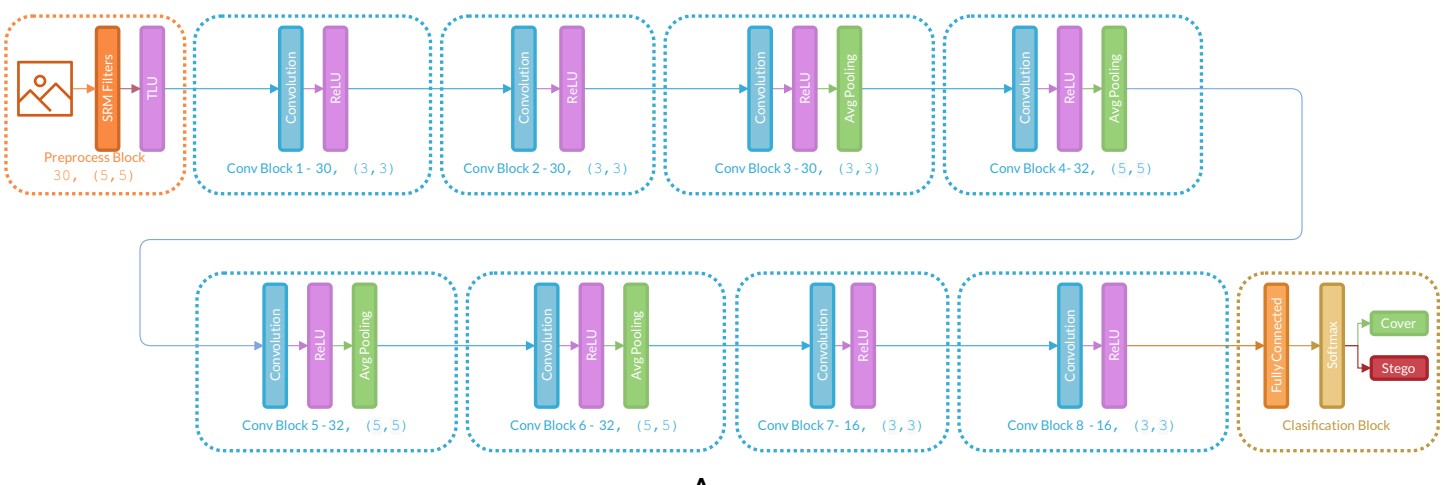

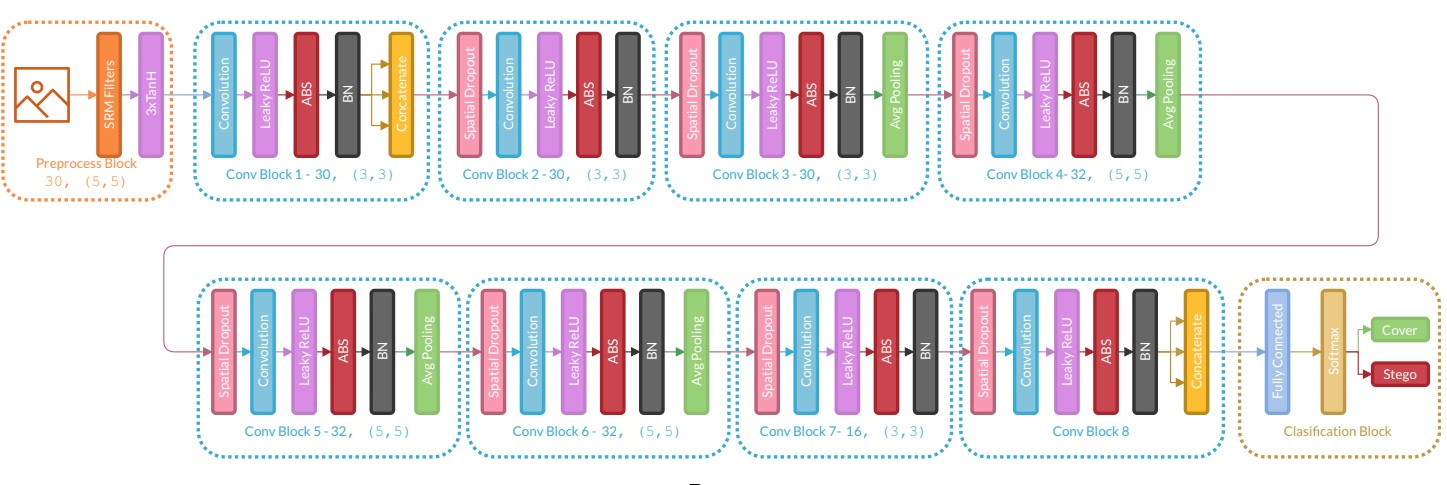

**Figure 3 (A) Architecture of the original Ye-Net, (B) Ye-Net architecture with the strategy applied.**

gradient descent optimizer momentum is 0.95, and a learning rate initialized to 0.005. Finally, activation in convolutional layers was a modified ReLU with a negative slope of 0.1, converting a ReLU into a Leaky ReLU.

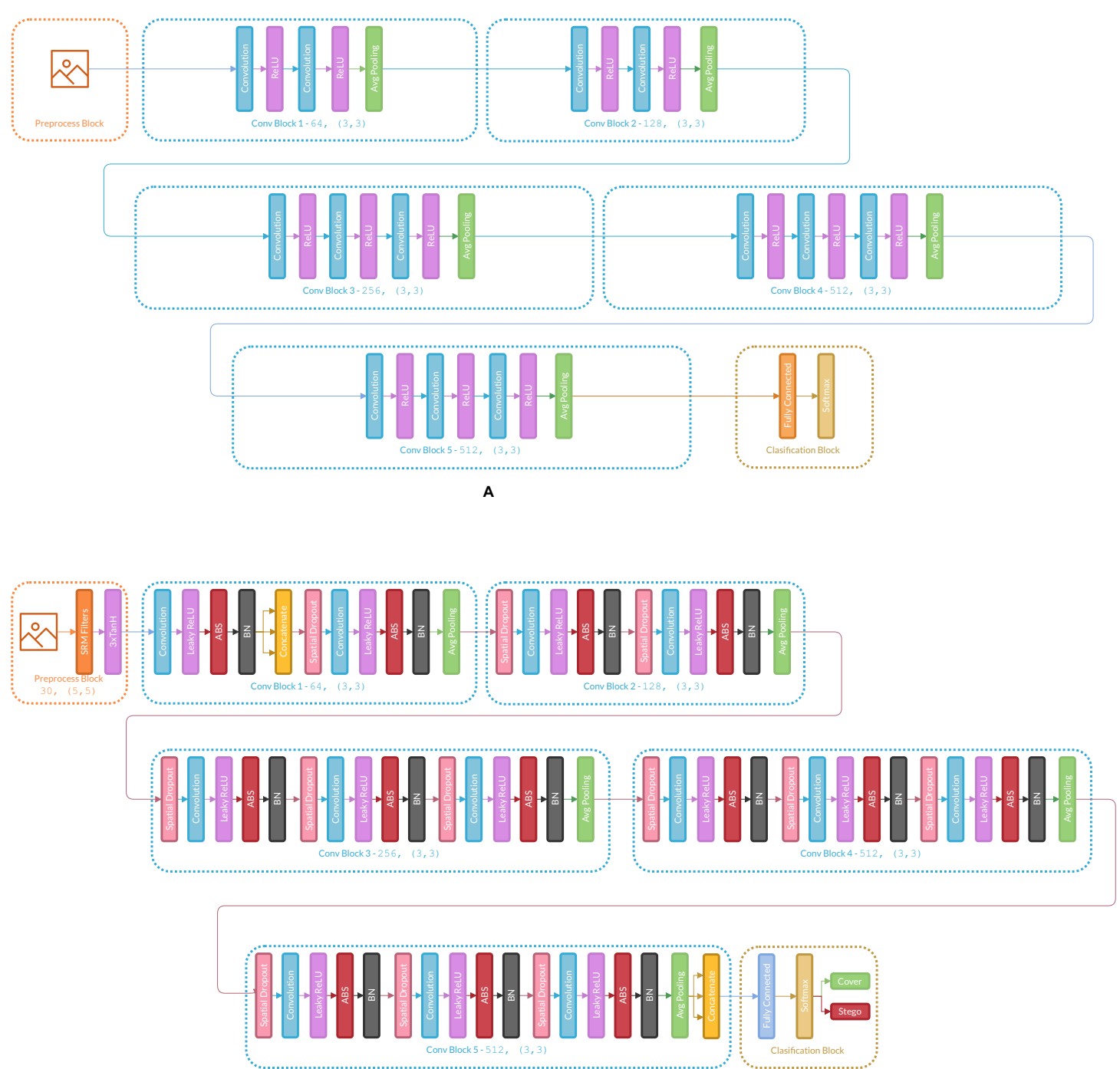

**Figure 4** (A) Architecture of the original VGG16, (B) VGG16 architecture with the strategy applied.

## Training

The batch size is set to 64 images for the steganalysis CNNs (Xu-Net, Ye-Net, and Yedroudj-Net) and 32 images for the VGG16 and VGG19 due to their bigger network size. The number of epochs needed to train the architectures varies depending on the database,

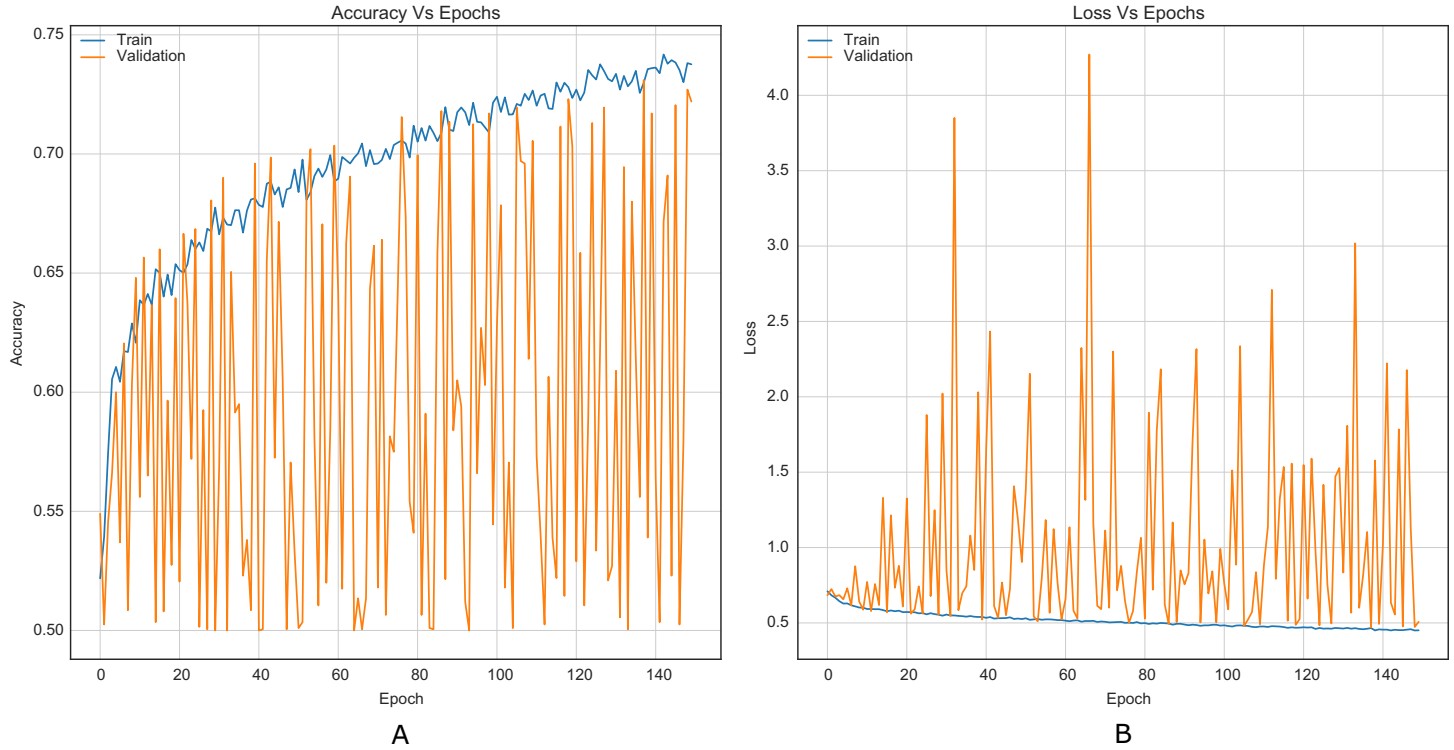

**Figure 5 Training curves of Xu-Net with BOSSBase 1.01 S-UNIWARD 0.4 bpp without strategy. (A) Accuracy, (B) loss.**

payload, and model. Ye-Net and Yedroudj-Net are trained for 100 epochs in both databases with 0.4 bpp, while Xu-Net was trained 150 epochs. VGG16 and VGG19 in BOSSBase 1.01 with 0.4 bpp are trained for 100 and 160 epochs, respectively; in BOSSBase 1.01 + BOWS 2 with 0.4 bpp, only 60 epochs are necessary for convergence in both networks, continuing the training of the model trained only in BOSSBase 1.01 dataset. To train the networks with 0.2 bpp, their weights were initialized with the weights obtained from the model trained with 0.4 bpp (transfer learning). For 0.2 bpp, all CNNs were trained for 50 epochs.

## Software and hardware

Most of the architectures and experiments implementations used Python 3.8.1 and TensorFlow (*Abadi et al., 2015*) 2.2.0 in a workstation running Ubuntu 20.04 LTS as an operating system. The computer runs a GeForce RTX 2080 Ti (11 GB), CUDA Version 11.0, an AMD Ryzen 9 3950X 16-Core Processor, and 128 GB of RAM. The rest of the implementations used the Google Colaboratory platform in an environment with a Tesla P100 PCIe (16 GB), CUDA Version 10.1, and 25.51 GB of RAM.

## RESULTS

For comparison purposes, all CNNs were implemented as presented in the original papers. This allowed to establish a baseline along with results reported in literature. Figure 5 shows the training curves for Xu-Net without the strategy.

**Table 1 Accuracy in test S-UNIWARD stego-images with different payloads using BOSSBase 1.01 and BOSSBase 1.01 + BOWS 2.** The bold entries indicate the results obtained with the strategy application on the CNN's. These results show the accuracies on test data for 0.2 bpp and 0.4 bpp.

| Dataset | BOSSBase 1.01 | | | | BOSSBase 1.01 + BOWS 2 | | | |
|---|---|---|---|---|---|---|---|---|
| Results | Reported in literature | | Strategy | | Reported in literature | | Strategy | |
| Payload | 0.2 bpp | 0.4 bpp | 0.2 bpp | 0.4 bpp | 0.2 bpp | 0.4 bpp | 0.2 bpp | 0.4 bpp |
| Xu-Net | 0.6090 | 0.7280 | **0.6829** | **0.7819** | – | – | **0.7121** | **0.8182** |
| Ye-Net | 0.6000 | 0.6880 | **0.7103** | **0.8101** | – | – | **0.7269** | **0.8338** |
| Yedroudj-Net | 0.6330 | 0.7720 | **0.6773** | **0.7964** | 0.6560 | – | **0.7335** | **0.8415** |

**Table 2 Accuracy in test WOW stego-images with different payloads using BOSSBase 1.01 and BOSSBase 1.01 + BOWS 2.** The bold entries indicate the results obtained with the strategy application on the CNN's. These results show the accuracies on test data for 0.2 bpp and 0.4 bpp.

| Dataset | BOSSBase 1.01 | | | | BOSSBase 1.01 + BOWS 2 | | | |
|---|---|---|---|---|---|---|---|---|
| Results | Reported in literature | | Strategy | | Reported in literature | | Strategy | |
| Payload | 0.2 bpp | 0.4 bpp | 0.2 bpp | 0.4 bpp | 0.2 bpp | 0.4 bpp | 0.2 bpp | 0.4 bpp |
| Xu-Net | 0.6760 | 0.7930 | **0.7352** | **0.8221** | – | – | **0.7483** | **0.8476** |
| Ye-Net | 0.6690 | 0.7680 | **0.7547** | **0.8451** | 0.7390 | – | **0.7713** | **0.8623** |
| Yedroudj-Net | 0.7220 | 0.8590 | **0.7623** | **0.8470** | 0.7630 | – | **0.7822** | **0.8691** |

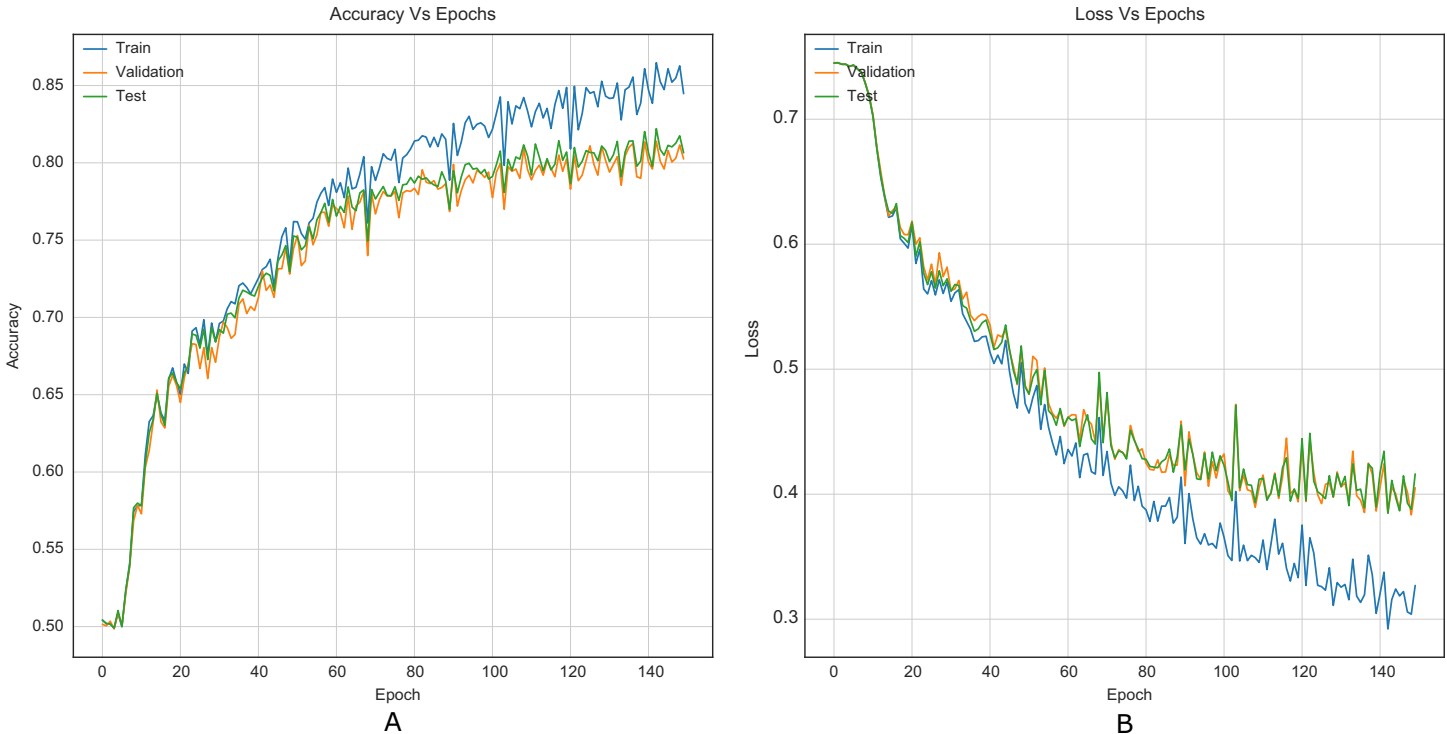

**Figure 6 Training curves of Xu-Net with BOSSBase 1.01 WOW 0.4 bpp. (A) Accuracy, (B) loss.**

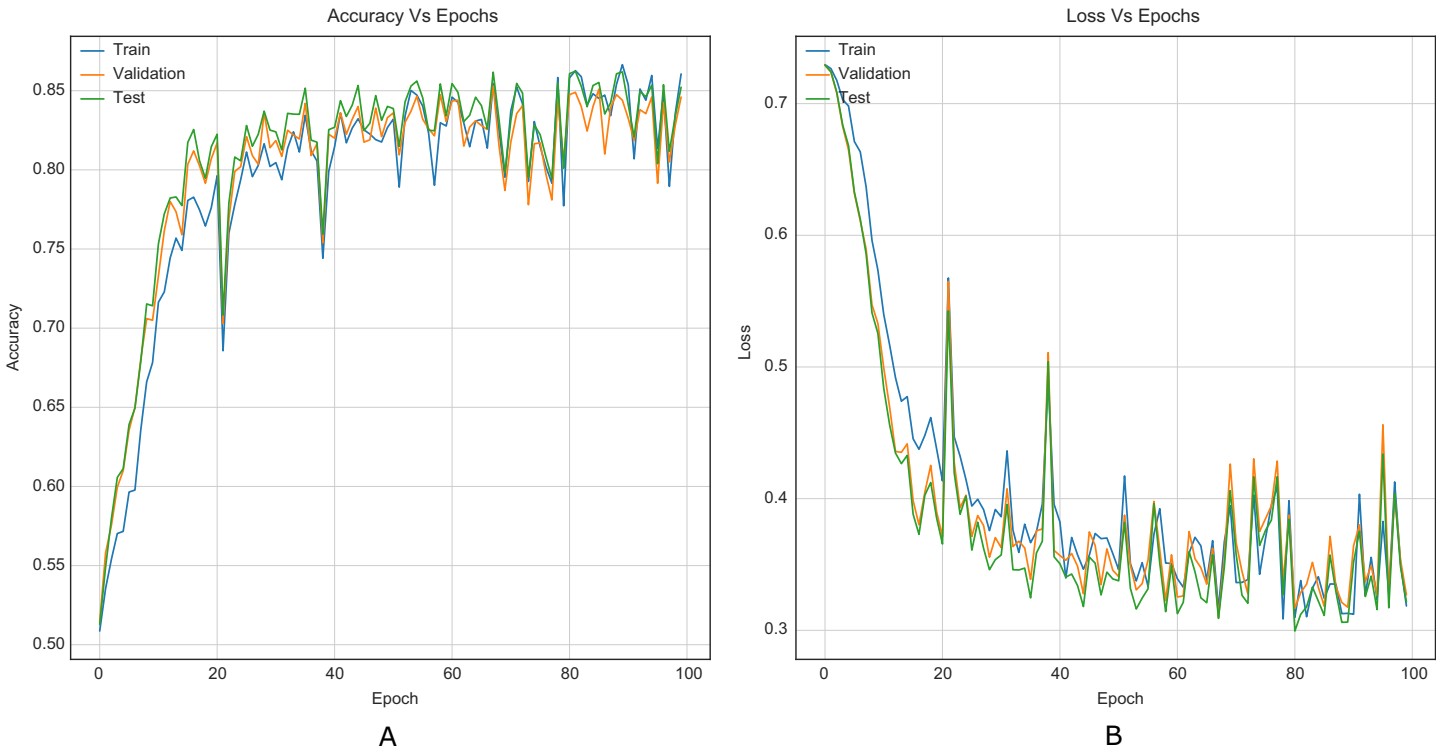

**Figure 7 Training curves of Ye-Net with BOSSBase 1.01 + BOWS 2 WOW 0.4 bpp. (A) Accuracy, (B) loss.**

The strategy proposed for CNNs was trained and tested using the images (cover and stego) of BOSSBase 1.01 and BOSSBase 1.01 + BOWS 2, with steganographic algorithms S-UNIWARD and WOW at 0.2 and 0.4 bpp. To evaluate performance, we use the highest accuracy in testing. The arrangement with the BOSSBase 1.01 and BOSSBase 1.01 + BOWS 2 databases in S-UNIWARD is recorded in Table 1 and with WOW in Table 2. The obtained accuracy and loss curves of CNN Xu-Net in WOW with 0.4bpp BOSSBase 1.01 are presented in Fig. 6, Ye-Net in WOW 0.4bpp with BOSSBase 1.01 + BOWS 2 in Fig. 7, and Yedroudj-Net in S-UNIWARD 0.4bpp with BOSSBase 1.01 are presented in Fig. 8, to corroborate the operation of the strategy.

According to the results listed in Tables 1 and 2, the strategy for CNNs helps to overcome the reported accuracies in state-of-the-art (*Reinel, Raul & Gustavo, 2019*; *Ye, Ni & Yi, 2017*; *Yedroudj, Comby & Chaumont, 2018*), in WOW and S-UNIWARD with 0.2 and 0.4 bpp payloads. Through the strategy, VGG16 and VGG19 classification networks were transformed into steganalysis networks (VGG16Stego and VGG19Stego see Fig. 4), demonstrating that by adding the strategy to classification CNNs, they become optimal for steganalysis. Table 3 recorded the accuracy of VGG16Stego and VGG19Stego with the BOSSBase 1.01 and BOSSBase 1.01 + BOWS 2 databases in S-UNIWARD and Table 4 in WOW. In those tables, Max Pooling and Average Pooling are variants of model results. Average Pooling is more used in steganalysis models to obtain the steganographic

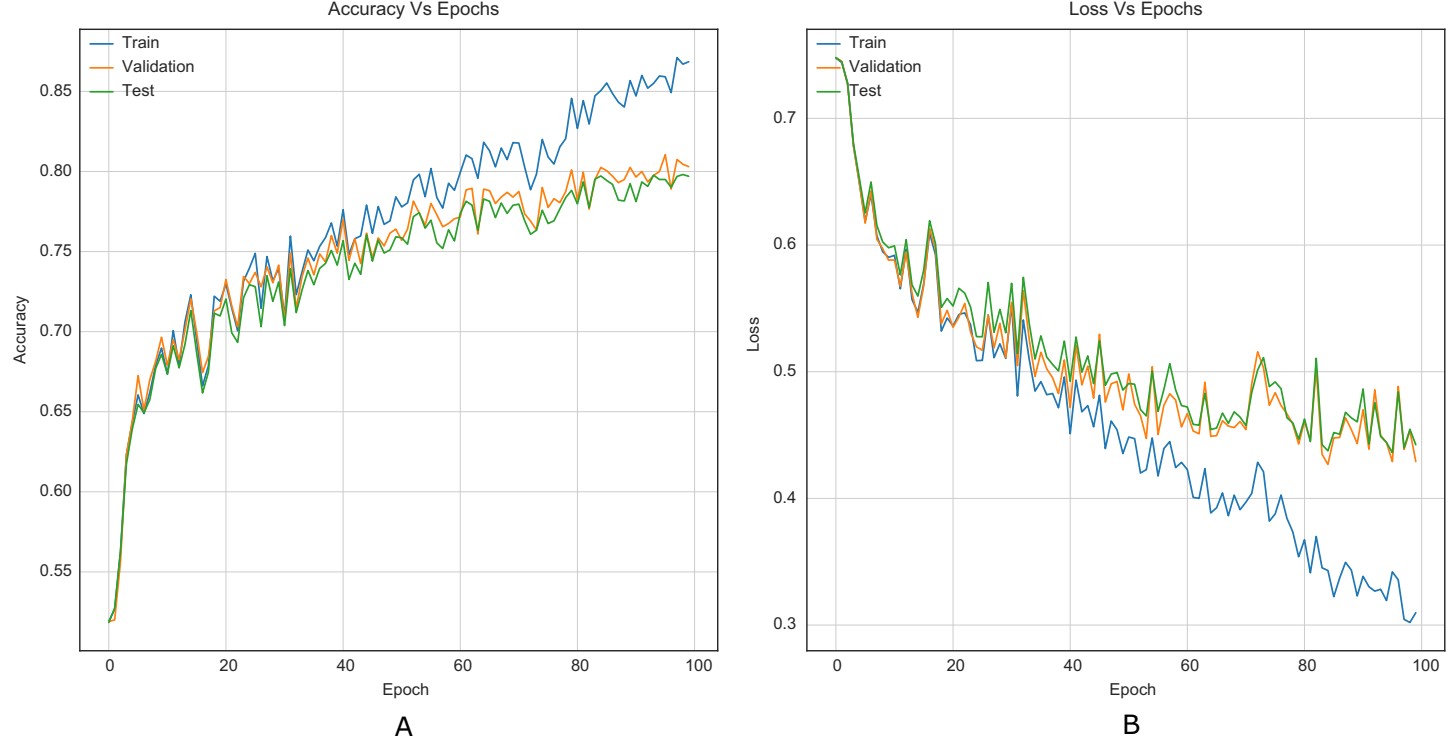

**Figure 8** Training curves of Yedroudj-Net with BOSSBase 1.01 S-UNIWARD 0.4 bpp. (A) Accuracy, (B) loss.

**Table 3** Accuracy in test S-UNIWARD stego-images with VGG16Stego-VGG19Stego and different payloads using BOSSBase 1.01 and BOSSBase 1.01 + BOWS 2.

| Dataset | BOSSBase 1.01 | | | | BOSSBase 1.01 + BOWS 2 | | | |
|---|---|---|---|---|---|---|---|---|
| Pooling | Max pooling | | Average pooling | | Max pooling | | Average pooling | |
| Payload | 0.2 bpp | 0.4 bpp | 0.2 bpp | 0.4 bpp | 0.2 bpp | 0.4 bpp | 0.2 bpp | 0.4 bpp |
| VGG16Stego | 0.7370 | 0.8291 | 0.7356 | 0.8370 | 0.7513 | 0.8545 | 0.7473 | 0.8511 |
| VGG19Stego | 0.7420 | 0.8210 | 0.7417 | 0.8291 | 0.7409 | 0.8520 | 0.7550 | 0.8490 |

**Table 4** Accuracy in test WOW stego-images with VGG16Stego-VGG19Stego and different payloads using BOSSBase 1.01 and BOSSBase 1.01 + BOWS 2.

| Dataset | BOSSBase 1.01 | | | | BOSSBase 1.01 + BOWS 2 | | | |
|---|---|---|---|---|---|---|---|---|
| Pooling | Max pooling | | Average pooling | | Max pooling | | Average pooling | |
| Payload | 0.2 bpp | 0.4 bpp | 0.2 bpp | 0.4 bpp | 0.2 bpp | 0.4 bpp | 0.2 bpp | 0.4 bpp |
| VGG16Stego | 0.7760 | 0.8556 | 0.7857 | 0.8640 | 0.8059 | 0.8825 | 0.8017 | 0.8830 |
| VGG19Stego | 0.7820 | 0.8570 | 0.7930 | 0.8656 | 0.8060 | 0.8833 | 0.8055 | 0.8857 |

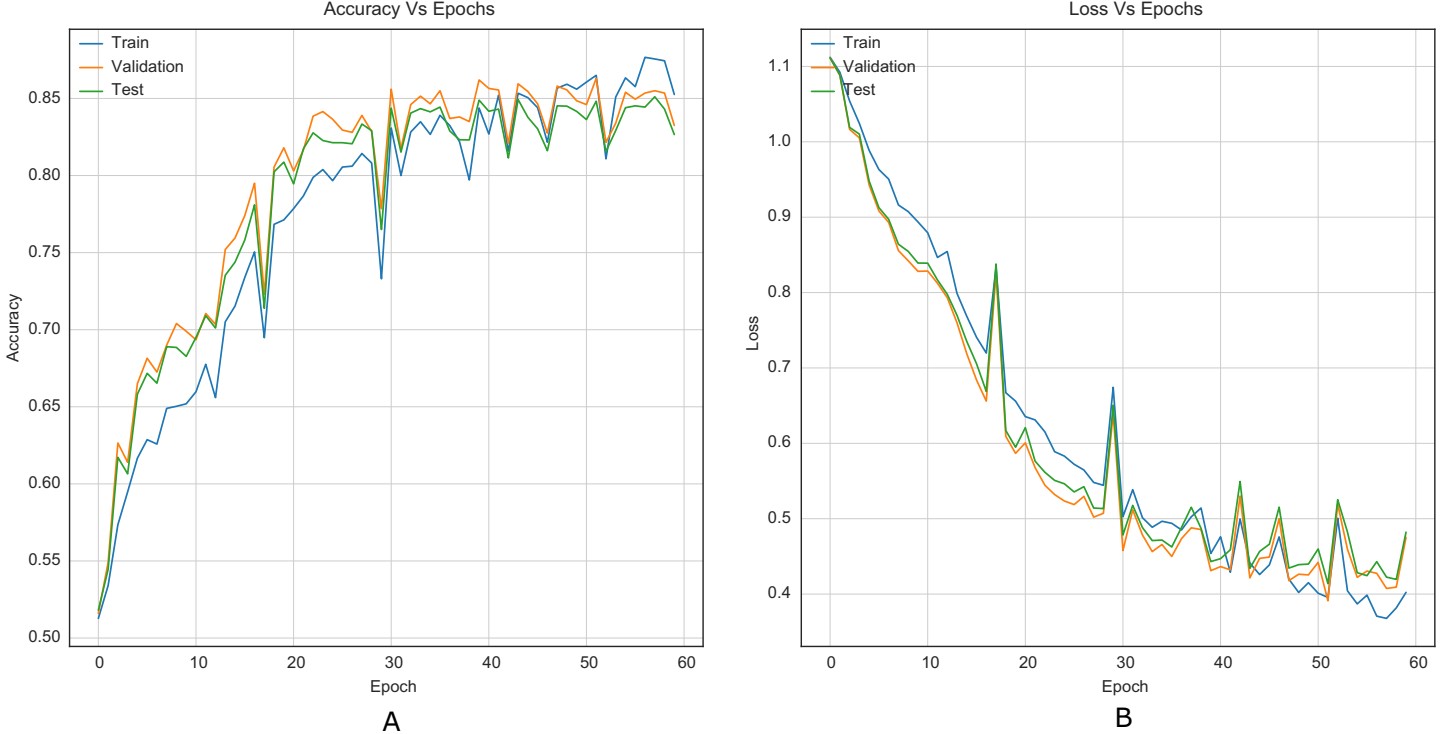

**Figure 9** Training curves of VGG16Stego Average Pooling with BOSSBase 1.01 + BOWS 2 S-UNIWARD 0.4 bpp. (A) Accuracy, (B) loss.

noise of the images. Max Pooling is used in the classification CNNs (VGG16 and VGG19) to get the most relevant image characteristics. The CNN VGG19Stego accuracy and loss curves were obtained in WOW at 0.4bpp with BOSSBase 1.01, VGG16Stego with Average Pooling in S-UNIWARD 0.4bpp BOSSBase 1.01 + BOWS 2 are presented in Fig. 9, to corroborate the operation of the strategy.

Figure 10 presents the ROC curves of all experiments in S-UNIWARD, while Fig. 11 shows ROC curves in the WOW algorithm.

## DISCUSSION

This study presents the results of testing different combinations of computational elements and hyper-parameters of CNN architectures applied to image steganalysis in the spatial domain, which led to identifying relevant elements for this task and the designing a general strategy to improve the CNNs. There were improvements in the convergence and stability of the training process and steganographic images' detection accuracy.

Regarding detection accuracy, the steganalysis CNNs (Xu-Net, Ye-Net, and Yedroudj-Net) perceived an improvement from 2% up to 10% in both steganographic algorithms and payloads. This performance boost can be attributed to the preprocessing stage involving the SRM filter bank and the modified 3 × TanH activation function. The SRM filter bank objective is to enhance steganographic noise in images, and as proven before, it

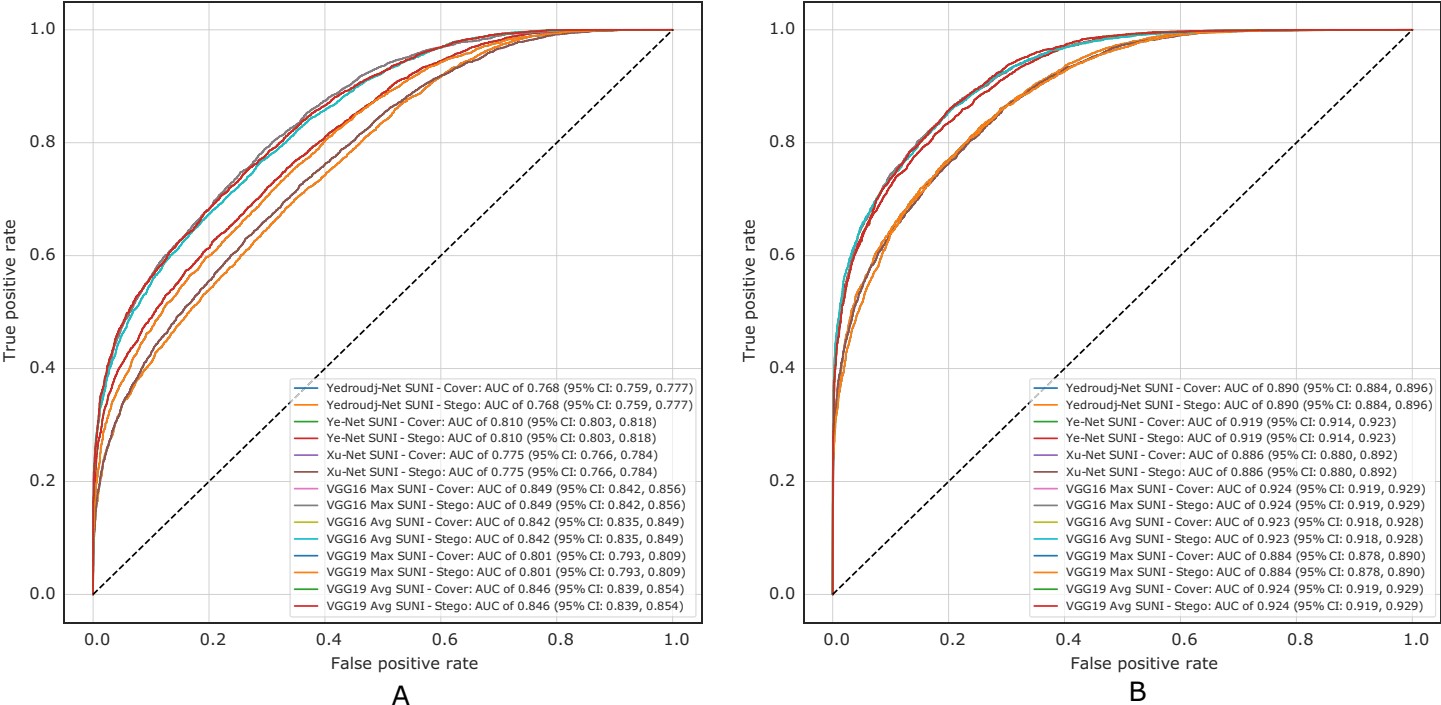

**Figure 10 ROC test curves of all experiment in S-UNIWARD. (A) Payload of 0.2 bpp, (B) payload of 0.4 bpp.**

improves detection accuracy (*Ye, Ni & Yi, 2017*; *Yedroudj, Comby & Chaumont, 2018*). TLU function inspired the activation function. As shown by *Ye, Ni & Yi (2017)*, it is better at capturing the steganographic noise than other activation functions, and the threshold value that yielded the best results was $T = 3$. Both TLU and TanH have a similar shape, but the latter is a smooth differentiable function, and the amplification by 3 mimics the desired behavior of the TLU function. On the other hand, the VGG16 and VGG19 image classification CNNs did not surpass the 0.5 detection accuracy before applying the strategy. In contrast, with the strategy, the results overcome those achieved by the steganalysis CNNs. From the results presented in Tables 3 and 4, it is important to note that, in most cases, the results with Average Pooling are better than those achieved with Max Pooling. In general, Average Pooling is preferred in steganalysis applications because it preserves the steganographic noise better than Max Pooling *Qian et al. (2015)*, given its low amplitude compared to image content. Additionally, the three-layer classification stage provides deeper processing of the features extracted in the convolutional layers, improving detection accuracy.

The mentioned improvements in convergence refer to the lower number of epochs and iterations needed to train the CNNs, which means training in less time. In comparison, in the original paper of the Xu-Net (*Xu, Wu & Shi, 2016b*), it was trained for 120,000 iterations with a mini-batch of size 64; With the strategy, the Xu-Net architecture was trained for 18,750 iterations with the same mini-batch size, while

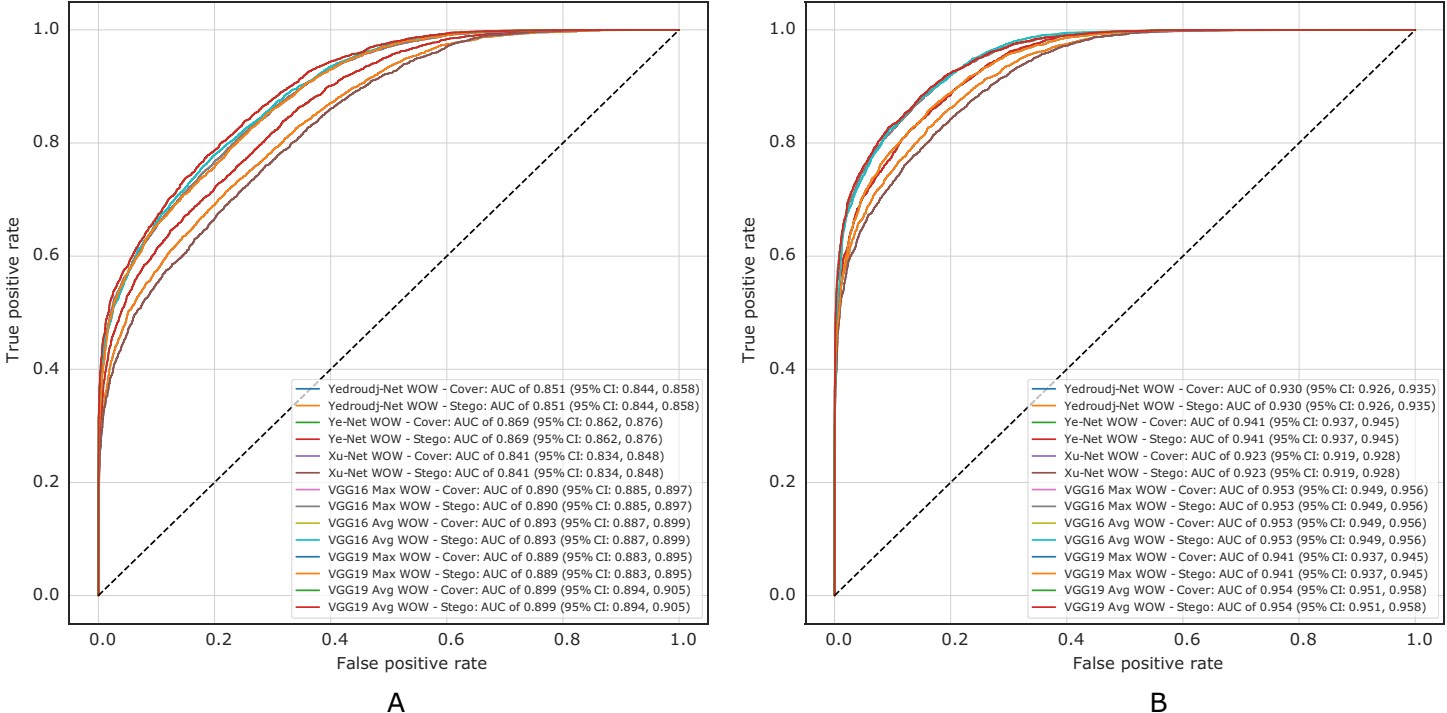

**Figure 11 ROC test curves of all experiments in WOW. (A) Payload of 0.2 bpp, (B) payload of 0.4 bpp.**

**Table 5 Approximate time of training in all models.**

| Dataset | BOSSBase 1.01 | | BOSSBase 1.01 + BOWS 2 | |
|---|---|---|---|---|
| Payload | 0.2 bpp | 0.4 bpp | 0.2 bpp | 0.4 bpp |
| Xu-Net | ~20 min | ~60 min | ~70 min | ~210 min |
| Ye-Net | ~80 min | ~180 min | ~140 min | ~280 min |
| Yedroudj-Net | ~100 min | ~220 min | ~350 min | ~400 min |
| VGG16Stego | ~180 min | ~240 min | ~400 min | ~460 min |
| VGG19Stego | ~200 min | ~310 min | ~410 min | ~580 min |

improving the detection accuracy. The training process duration can not be compared because they depend on other factors like hardware specifications. However, it is worth mentioning that it did not take longer than 10 h to train image classification CNNs and less than 6 h to train steganalysis CNNs. Table 5 is the approximated time of each CNN.

Similarly, it is challenging to compare training stability improvement, which refers to less variability on the training curves, given the original papers' lack of training curves. For this purpose, we were able to reproduce the original Xu-Net architecture to compare the training curves with and without the strategy. By comparing Figures 6 to 5, it is possible to observe how accuracy and loss curves vary less over time. In practice, the computational element found to improve the training stability was the Spatial Dropout.

By adding this operation before the convolutional layers, the training curves were smoother, although it also forces add epochs to reach convergence.

## CONCLUSIONS

This work presents a strategy to improve the CNNs applied to image steganalysis in the spatial domain. The strategy's key is combining the following computational elements: SRM filter bank and $3 \times$ TanH activation for the preprocessing stage, Spatial Dropout, Absolute Value layer, Batch Normalization and fully connected. The performance improvement can be seen in the convergence and stability of the training process and the detection accuracy. VGG16Stego and VGG19Stego obtained the best performances. Future work should be aimed to optimize the strategy and test it on recent steganalysis CNNs, SR-Net by *Boroumand, Chen & Fridrich (2018)* and Zhu-Net by *Zhang et al. (2019)*. Additionally, demonstrate experimentally the influence of each layer and hyperparameter added by the strategy.

### Funding

This work was supported by Universidad Autónoma de Manizales, Manizales, Colombia, under project No. 645-2019 TD. The funders had no role in study design, data collection and analysis, decision to publish, or preparation of the manuscript.

### Grant Disclosures

The following grant information was disclosed by the authors:
Universidad Autónoma de Manizales, Manizales, Colombia: 645-2019 TD.

### Competing Interests

The authors declare that they have no competing interests.

### Author Contributions

- Reinel Tabares-Soto conceived and designed the experiments, analyzed the data, authored or reviewed drafts of the paper, and approved the final draft.
- Harold Brayan Arteaga-Arteaga performed the experiments, analyzed the data, prepared figures and/or tables, authored or reviewed drafts of the paper, and approved the final draft.
- Alejandro Mora-Rubio performed the experiments, performed the computation work, prepared figures and/or tables, authored or reviewed drafts of the paper, and approved the final draft.
- Mario Alejandro Bravo-Ortíz performed the experiments, analyzed the data, prepared figures and/or tables, authored or reviewed drafts of the paper, and approved the final draft.
- Daniel Arias-Garzón performed the experiments, performed the computation work, prepared figures and/or tables, authored or reviewed drafts of the paper, and approved the final draft.

- Jesús Alejandro Alzate Grisales performed the experiments, prepared figures and/or tables, authored or reviewed drafts of the paper, and approved the final draft.
- Alejandro Burbano Jacome performed the experiments, performed the computation work, prepared figures and/or tables, authored or reviewed drafts of the paper, and approved the final draft.
- Simon Orozco-Arias conceived and designed the experiments, authored or reviewed drafts of the paper, and approved the final draft.
- Gustavo Isaza conceived and designed the experiments, authored or reviewed drafts of the paper, and approved the final draft.
- Raul Ramos Pollan conceived and designed the experiments, authored or reviewed drafts of the paper, and approved the final draft.

## Data Availability

Code and data are available at GitHub: https://github.com/BioAITeam/Strategy-to-improve-CNN-applied-to-digital-image-steganalysis-in-the-spatial-domain.

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
