# Peer review of "Strategy to improve the accuracy of convolutional neural network architectures applied to digital image steganalysis in the spatial domain"

_PeerJ Computer Science, doi:10.7717/peerj-cs.451_

## Round 0.1 · original submission · Minor Revisions

Both reviewers would like the authors to improve the paper and clarify the concerns raised.

·

Basic reporting

The writing of this article is particularly good to get what readers are concerned about. Thus, a positive recommendation would be provided under good expectations.

Experimental design

The authors divide the images using fixed sizes to create two databases. The partition of the database should reference the popular way in the deep learning community as well. For example, creating each corresponding steganographic image for each cover image and structuring it as a complete database, then allowing the designed algorithm to select appropriate proportions. Besides, giving a clear name to illustrate the combination of BOSSBase V1.01 and BOWS2 databases. Better to make a website based on GitHub to introduce the summarized databases.

Validity of the findings

The presented innovation meets the requirements that obtain a pleasant recommendation. Better to convert the current study to a benchmark for other researchers. Also, the strategy enclosing special elements might lack a scientific basis. Most of them might be empirical and there are difficulties to learn systematically. Of course, true actions depend on the authors.

Additional comments

Dear authors, so glad to read your paper regarding investigating the CNN (convolutional neural network) based methods applied to steganalysis and adding special modules to enhance the resultant accuracy.

Reviewer 2 ·

Basic reporting

no comment

Experimental design

no comment

Validity of the findings

Novel enough for publication.

Additional comments

The authors propose a general strategy to improve existing deep learning based steganalytic methods. The strategy makes modifications in the stages of pre-processing, feature extraction, and classification using SRM filters, batch normalization, absolute value layer, and spatial dropout. The proposed scheme is novel and interesting. The manuscript is suitable for publication. Some comments are as follows:
- The proposed scheme makes modifications in several stages of deep steganalysis. The effectiveness of modifications in each stage should be discussed. For example, some experimental results should be given.
- It is better to summarize the contributions of the manuscript in the end of Introduction.

---

## Round 0.2 · accepted · Accept

Thanks for the revision. The paper is ready to be accepted.